# Sustainable Return to Work for Workers with Mental Health and Musculoskeletal Conditions

**DOI:** 10.3390/ijerph20021057

**Published:** 2023-01-06

**Authors:** Abasiama Etuknwa, Kevin Daniels, Rachel Nayani, Constanze Eib

**Affiliations:** 1Swiss Reinsurance Company Ltd., London EC3A 8EP, UK; 2Norwich Research Park, Norwich Business School, University of East Anglia, Norwich NR4 7TJ, UK; 3Department of Psychology, Uppsala University, 752 36 Uppsala, Sweden

**Keywords:** flexible working, line manager, rehabilitation, return to work, sickness absence management, working conditions-job quality

## Abstract

Common mental health and musculoskeletal disorders (CMDs and MSDs) are two of the most significant causes of non-participation in employment amongst working age adults. Background: This case study fills an important gap in the scientific literature on reintegration back to work after sickness absence due to CMDs and MSDs. It particularly examines the return to work (RTW) experiences of sick-listed employees to understand the facilitators and barriers of sustainable RTW. Methods: Using a realist evaluation approach within a qualitative inquiry, perceptions of employees were explored to provide in-depth understanding of what, how and under what circumstances sustainable RTW can be enabled for employees absent on a short- or long-term basis. Repeat face-to-face semi-structured interviews were conducted with 22 participants (15 women and 7 men, aged 30–50 years and sick-listed with MSDs and CMDs) who were recruited using purposive sampling. Data was thematically analysed. Results: A total of 2 main codes and 5 subcodes were developed and grouped into three theoretical abstractions. As a result of validating the context, mechanism, and outcome configurations with accounts of participants, all three initial theories explaining the most prominent mechanisms that either facilitates or impedes a sustainable RTW for people with CMDs and MSDs were justified. Conclusions: Our findings reveal the active role of line managers on the RTW outcomes of returning employees. However, line-manager’s competence and ability to effectively support and implement appropriate RTW strategies suited to employees’ hinges on working in alignment with key stakeholders and returning employees.

## 1. Introduction

Despite a range of policies, guidelines and best practices put in place by workplaces and governments to reduce days lost to sickness absence as a result of common mental (e.g., anxiety, depression) and musculoskeletal disorders (CMDs and MSDs), high absence rates due to these conditions persist [1,2,3], On the one hand, CMDs represents the most reported mental health problems which include stress, anxiety, and depression [4]. According to Health and Safety Executive (HSE) [4], they are often a response to a traumatic life event or experiences in a person’s work life, home life or a combination of both which impacts their ability to function. On the other hand, MSDs is an impairment in the muscles, bones, joints, and adjacent connective tissues that is typically characterised by pain (often persistent) and limitations in mobility and dexterity, reducing people’s ability to work and participate in society [5]. These conditions can be chronic with high levels of relapse [6], leaving workers at higher risk of premature exit from the labour market [7,8] and at greater risk of becoming marginalised [9,10], making the case for understanding the effective ways to help sick-listed employees return to work (RTW) stronger. In January 2022, 1.8 million workers in the UK had a work-related illness [11]. Of these 1.8 million workers, 914,000 were absent for CMDs and 477,000 for MSDs [11,12]. These yearly significant incline in absent rate for these conditions may suggest a gap in effective implementation strategies for better supporting workers back to work. It is for this reason that we are particularly focusing on CMDs and MSDs in this study. We believe that both employers and government will benefit from findings from this study to reduce absence rates and cost on absence.

RTW is important because employment provides employees with positive outcomes such as self-esteem, social connection, and financial wellbeing [13,14]. However, there are tensions between what employees need for reintegration into workplaces and what employers need for rapid return to the workplace [15,16,17]. Furthermore, sick-listed workers are the least powerful actors but would profit the most from successful RTW, because of health and financial benefits of working [15]. Moreover, given the impact of the COVID-19 pandemic on risk factors for and incidence of MSDs and CMDs, e.g., associated with remote working and working with unsuitable equipment [18], understanding workplace factors relevant to sustainable RTW has great prominence, especially when these factors are a product of a larger systemic/organisational issue [19].

The process of recovery from ill-health is accelerated by early RTW [20], yet the process of returning to work for employees sick-listed with CMDs and MSDs is complex involving multiple interacting factors and the cooperation of different stakeholders [21,22,23]. Ideally, when employees are returning to work after a period of sick leave, they follow a planned RTW process that consists of a series of evolving phases which takes account of treatment plans, state of recovery and employee capabilities at the point of return [24]. However, in many cases, due to the complexity of the RTW process, a proportion of these sick-listed employees experience a variable and often undesirable RTW course including extended or intermittent absence that results in significant individual, employer, and societal costs [25]. Therefore, poor implementation of the RTW process could result in unanticipated RTW outcomes, which may explain why agreed RTW strategies do not always have the intended effect of benefiting sick-listed employees [25]. In Jetha et al.’s study [25], while RTW strategies appeared to benefit some sick-listed workers, minimal effects were observed in others. These inconsistencies in effects could suggest that there may have been variabilities in the competency of the RTW coordinators and the adequacy of the agreed RTW strategy for workers. It is therefore clear that implementation and competencies of key RTW actors are a fruitful area of inquiry in understanding how best to support returning workers.

The implementation of workplace RTW interventions have been investigated to a great extent [26,27,28], with some authors suggesting that the key determinants to successful RTW outcomes or job retention is provision of adequate adjustments or workplace accommodations and resolving work-related challenges [29,30,31]. However, there is still limited knowledge of quality workplace based RTW interventions outlining what agreed RTW strategies, how and under what circumstances they facilitate sustainable RTW outcomes for people with CMDs and MSDs [32]. In particular, findings from Martimo’s [33] and Viikari-Juntura et al.’s [34] studies show that simply providing such adjustments that may reduce the risk of aggravating employee’s on-going condition comes with difficulties, because of the psychosocial, workplace or management issues that may influence effects. Other authors suggest that the sustainability of RTW outcomes for people with CMDs and MSDs may be impacted by the length of sickness absence [35]. For instance, some authors linking length of sickness absence and RTW outcomes have focused on the likelihood of RTW for people sick-listed as either short-term or long-term absentees [36,37,38]. In such studies, compared to people classed as long-term absentees (defined as absence period lasting more than six weeks), sustainable RTW was found to be enhanced for people sick-listed on a shorter-term basis [6]. However, it is unclear if sustainable RTW was heightened for people classed as short-term absentees because of the suitability of components of implemented workplace strategies or because of other factors.

Therefore, the current study examines the workplace RTW implementation process and the competencies of key actors to identify the enablers of sustainable RTW for people with CMDs and MSDs. It also examines the specific strategies that benefit employees classed as short-term and long-term absentees.

## 2. Materials and Methods

A realist evaluation approach [39,40,41] was considered an appropriate design for this qualitative study given our focus on “how” and “why” questions, identify the contextual conditions relevant to the phenomenon under study and understand the boundaries between the phenomenon and the context [42]. Additionally, a realist evaluation approach enabled testing of initial theories/ideas across purposively selected cases [43] linked together through a common issue or issues [44].

### 2.1. Recruitment

Participants were recruited through contacts of the research team and purposively sampled from the UK public services industry to maximise rigour about the inclusion of target groups that would aid the ease of applicability and transferability instead of generalizability. Included participants where employees were returning or had returned to work after ill-health due to MSDs and/or CMDs. Participants were provided with packs containing an information sheet on the research and a consent form according to the ethical requirement of the university. The sample comprised 22 employees (15 women and 7 men), aged 30–50 years and sick-listed with MSDs and CMDs, recruited from a local county council and a further education institution. Eleven participants reported being absent for CMDs, eight participants for MSDs and three for both MSDs and CMDs (Appendix A, participants’ descriptions).

### 2.2. Realist Evaluation Phase

Data collection and analysis was conducted within 3 realist evaluation phases; phase 1 -theory gleaning, phase 2–theory refining or creation and phase 3–theory consolidation (See Appendix A).

#### 2.2.1. Phase 1–Theory Gleaning

Identifying theories on how a programme works in a realist evaluation can be articulated using a wide range of approaches to include reviews or expert panels [45]. However, Doi et al. [46] suggests that also engaging stakeholders is a useful way in unpacking realist evaluation theories. For these reasons, theories for this study were first gathered deductively from a thorough review of existing literature reporting on the effects of organizational factors on sustainable return to work, and then inductively from inferences from managers who are responsible for the RTW process. As the review identified workplace support as the most consistent evidence, the interview with line managers line managers highlighted good quality RTW, workload clarity and workplace support as important factors underpinning sustainable RTW. See Appendix A for a detailed report of the theory gleaning phase.

##### Data Collection

In line with critical realist evaluation methodology, the interview questions focused on the intervention context, mechanisms, and outcomes (CMO); data gathering, and analysis was an iterative process as initial theorisation was gleaned and refined [45]. Initial CMO configurations around factors that facilitate or impede sustainable RTW were gleaned from two sources: a systematic review of the RTW literature and interviews with four line-managers who manage the RTW process (Appendix A). Through a process of retro-duction and theoretical abstraction, we developed initial RTW theories using the CMO configuration structure. Within this approach, each mechanism was linked with the observed outcomes and then the context in which the mechanisms were dependent on was assessed to establish its link with the RTW process and the actors involved. After obtaining the inferred CMO configuration, abductive reasoning was applied to identify the possible explanatory CMO configuration and obtain testable hypothesis of the initial RTW theory [47] (Appendix A). Interviews were recorded and verbatim transcribed. Insights obtained from both the review and managers’ interviews informed the design of the employee interview guide (Appendix A). Repeat face-to-face semi-structured interviews were conducted with 20 sick-listed participants (42 interviews in total; 2 missed interviews were due to resignations). The second interviews clarified and refined the emerging theorization.

#### 2.2.2. Phase 2–Theory Refining or Creation

In this phase, data were analysed, and emerging theories from the data aided comparison with initial theories.

##### Data Analysis

The analytical method incorporated both the data-driven inductive approach and theory-driven deductive approaches [48]. Initial CMO configurations were tested and refined by analysing interview data and generated ideas or links between or among phenomena were consolidated through a process of theoretical abstraction [49]. This allowed a detailed and rich exploration of how RTW is facilitated for employees [50].

Interview data were coded thematically using NVivo software and according to an initial coding framework (Appendix A). Where data did not fit, new codes were created via open coding and refined and/or merged where sharing properties and dimensions [51] through constant comparison. In the theory consolidation stage, we used a process of retroduction and theoretical abstraction to look at how the CMO configurations of the developed theories came together as a whole [47] (Appendix A). We reassembled the data to weave the story back within cases and compared across cases [51,52] to identify dominant explanatory conditions. Through the constant comparison and iteration to both data and the literature the findings presented here offer the most empirically robust explanations on “how, why and under what circumstance” a sustainable RTW is either facilitated or impeded by organisational factors.

#### 2.2.3. Phase 3–Theory Consolidation

In phase three, final theories validated by participants in the second interviews and the theories more worthy of consideration were finally fine-tuned [45]. The CMO configuration of the theories were either refined (where necessary) or defined (in the case of a new theory) to capture the precise explanations around the factors that influence sustainable RTW for participants sick-listed with CMDs and MSDs.

## 3. Results

Codes and sub-codes were developed and grouped into three theoretical abstractions drawn from the theories (Figure 1). These theoretical abstractions highlight the most prominent mechanisms either facilitating or impeding RTW. They are:Choices presented to line managers during the RTW process.Ongoing dialogue between manager and employee on specific concerns/re-negotiation processes.The alignment of stakeholders informs the choices presented to line managers during the re-negotiation process.

### 3.1. Choices Presented to Line Managers during the RTW Process

The choices presented to line managers either enable or restrict their ability to implement an effective RTW strategy during the RTW process, that precedes sustainable RTW outcomes for returning employees. The main code under this theoretical abstraction is good quality RTW process. with sub codes relating to alignment of key stakeholders and a competent and supportive line manager.

#### 3.1.1. Good Quality RTW Process

Participants’ descriptions of their workplace RTW practices based on their experiences highlighted two main components of a good quality RTW process (sub-codes): the alignment of key stakeholders around RTW strategy and a competent and supportive line manager.

##### The Alignment of Key Stakeholders around RTW Strategy

According to all participants, the RTW interview/meeting is a mandatory process within the policy of the organisation, that is arranged by line managers with sick-listed employees as soon as they RTW. These meetings were aimed at determining employee’s stage of recovery, restrictions/limitations and their needs regarding what services or resources would help the RTW process go smoothly. Based on recommendations from the medical consultant/GP or occupational health service, a RTW strategy appropriate for returning worker was agreed upon in a next step.


*“When I came back, we had a return-to-work interview which went ok… Ok, so, I started with low level of cases and I had phased return that was suggested by Occupational Health”*
(015-F-40+)

In some cases, the presence of representatives from the human resources department, occupational health, and other services was required in the RTW meeting for the purpose of ensuring that the right course of action was implemented.

All participants agreed that a properly implemented RTW process plays a role in sustainable RTW outcomes.


*“It was me feeling confident that the manager understood me as a person and understood my condition… I think it helped them understand better what support they needed to put in place.”*
(022-M-40+)

Seventeen participants believed that the RTW strategy implemented was effective, particularly those who had a phased return or flexible working options. A cross-section of these participants showed that effects of RTW strategies varied across condition and length of absence (see Appendix A).

Across both organisations, the effectiveness of agreed accommodations seemed to be contingent on approval from senior management, for example,


*“….if I say, ‘my back’s bad, driving is typical, can we consider the working from home because let’s face it, you haven’t even answered me from January’ (Laughing). It will be… I will ask the service management and then I’m pretty certain the answer will be NO! So, it is kind of blocked. So, the line manager is lovely, she’ll listen to you, but they are very restricted in what they can do to help. So, 9 out of 10 times it’s blocked.”*
(002-F-30+)

Evidence shows that where provision of adequate accommodations for sick-listed employees is concerned, line managers are restricted in their supportive capacity. In other words, where accommodations were not approved for deployment, chances of successful return to work were reduced, therefore, increasing the likelihood of a relapse.

##### Competent and Supportive Line Manager

Four participants revealed that effective or ineffective management and implementation of adequate RTW strategies was dependent on having a competent and supportive line manager, for example,


*“…The manager who I was working with at the time during my return was far better equipped to deal with people in my position and people with some mental health issues. The manager back where I was in my substantive role had absolutely zero ability in my opinion and I think others as well deal with that kind of situation.”*
(022-M-40+)

Effective RTW strategies are not dependent on an organisation having the right plan in place, but more about managers being supportive and having the experience or competence in managing the RTW process. However, where line managers do not have the competence to handle or manage the RTW process, efforts at sustaining RTW may be futile. Evidence of the negative impact of poor management by an unsupportive line manager on RTW outcomes despite provision of a phased return or flexible working option was echoed across three participants sick-listed with CMDs (019-M-40+, 022-M-40+ and 015-F-400). According to them, RTW failed because the process was poorly managed and the felt unsupported throughout the process.

### 3.2. Ongoing Dialogue between Manager and Employee on Specific Concerns/Re-Negotiation Processes

This abstraction describes the on-going dialogue/re-negotiation processes between line managers and returning employees in agreeing on the most appropriate work accommodation. The main code under this abstraction relates to work accommodation with two subcodes: work accommodation as an on-going dialogue, workload clarity and hindrances to re-negotiation.

#### 3.2.1. Work Accommodation

This code describes the reasonable accommodations or adjustments to employee’s job to enable them to perform and advance in their role on RTW. Accounts of participants highlight three key factors (sub-codes) that either facilitate or restrict employees’ re-negotiation dialogues: work accommodation as an on-going dialogue/re-negotiation, workload clarity dialogues and hindrances to re-negotiation dialogues.

##### Work Accommodation as an On-Going Dialogue/Re-Negotiation Process

Accounts of participants revealed that on-going dialogue and re-negotiation on appropriate work accommodations varied across participants based on their absence duration. Across all participants, RTW strategies implemented included a phased return, flexible working options such as a change in job task or role and workstation adjustment or provision of workstation accessories specifically for people with MSDs.

Participants classed as short-term absentees benefitted from flexible working options.


*“So, I came back, worked here and that didn’t work terribly well and then they said you could work from home for a few days. That really helped until I could sort out how to get to work better.”*
(014-F-40+, MSDs & 2 weeks absence)

Apart from working from home, other flexible working options offered to participants absent for a short-term period included a few days off within the week, choice of working in the mornings or afternoons, light duties and half days while still within their full-time contract until employees felt recovered enough to handle their full contractual duties.


*“I think what was helpful was the fact that I could work shorter hours and I got to choose them. And what I mean by that was I was offered…. You know, if I was going for half a day, would I prefer to do the morning, or would I prefer to do the afternoon. Because some... for me I chose the mornings because I get up and do the school runs anyways and I found that come the afternoon I was sore… So, for me that was really helpful.”*
(012-F-30+, MSDs & 6 weeks absence)

In comparison, participants classed as long-term absentees were of the view that a phased return was beneficial. Components phased within this strategy included reduced hours, reduced days, reduced workload, change in job role or level. A combination of these phased components was implemented for participants with both MSDs and CMDs and were gradually built up within a 4–6-week period until full-time status was attained.


*“I was phased on 25% for …. I can’t really remember a 100%. 25% was for two or three weeks, and then it was 50% for about two or three weeks and then it went to 75% for a week and then full time. So, it was needed another block between 75 and 100% if that makes sense. Because that feels like a very big step, from 75% to 100. It’s not because it’s no bigger than the other steps, but it’s just when you’re coming back it feels like a big jump.”*
(005-F-40+)


*“… It was occupational health who were obviously involved and then the colleagues that’s just sort of a subcontracted situation. But they, if I recall correctly, they… I went and spoke to them a couple of times and they made recommendations around the phase return and then my manager then took that on board. It was phased in terms of numbers of days of the week. And also, I went and worked for a different Department for a while, so that I was able to sort of break myself back into the grass roots of things because there was an awful lot of changes going on. So, it was quite important that I started to pick up on what was required and how the role had shifted.”*
(009-M-40+)

Unlike flexible working options, a phased return allowed participants to start on non-contractual hours and gently increase these over the agreed period. According to participants this phased strategy was very effective as it afforded them the time to gently get back into the work mode after being absent for an extended period. Phasing allowed them to get back in touch with how things work and become familiar with any operational changes that occurred during their absence.


*“I think I was off for six or seven months, so to come back in a couple of days a week to start off with you know, … I think it was a couple of days a week to start off with, and then that built over say six weeks back up to a full full-time role. It was… it made life easy… how can I describe? It meant that I didn’t feel that I was under immediate pressure to perform, to take on board everything that was going on.”*
(009-M-40+, CMD & 7 months)

All participants stated that a phased return has a higher tendency of working effectively if implemented in a supportive capacity and with better communication with the employee throughout the RTW process.

##### Workload Clarity

Ten participants’ perceptions around workload were considered alongside poor RTW outcomes. For some of these participants discussions around RTW did not address workload, leaving participants uncertain about what they could or could not do.


*“…it could be better simply by better communication and getting a clearer picture of what somebody can do when they come back rather than you go to occupational health, they say phased return, so your manager sits down with you and you work out the pattern of phased return and then off you go. Whereas, when you’re sitting down and talking about the phased return it needs to include ‘how are you emotionally, and physically what can you do?...”*
(016-F-40+)


*“… I suppose it’s that reassurance and almost reiteration of it. It’s very easy to just go ‘its ok, you can do this’ and then never mention it again. So, I think that repetition of ‘it’s still ok, this is ok, I accept that, this isn’t getting done but that’s ok’. But I think probably what I didn’t do with my manager was sit down and look at the work that I’ve got on because they probably don’t know the work that I’ve got, and they trust me to do that...”*
(020-M-40+)

The extract below captures the importance of ensuring the renegotiation dialogue is an on-going process, in this case when return to office was found to be untenable.


*“I think they were probably a bit surprised that I came in to work with my arm in a sling and tried to work (laughing). They kept on saying to me ‘are you sure you should be at work?’ And you know, ‘can you manage?’ And I thought I could manage. And I remember telling you this and then I realised that no, you know I couldn’t drive, and I had to walk back to the station with a laptop and another bag. And I just realised that I couldn’t do that on a daily basis because it was exhausting when you’re struggling the whole time and so then they said that I could work from home and that was what made a … that made a big difference.”*
(014-F-40+)

On-going dialogues accord line managers and employees the opportunity to evaluate the effectiveness of agreed accommodations to identify what works and what does not, and where necessary modify accommodations to suit employee’s initial and changing ill-health restrictions. The perception of these participants, therefore, stresses the importance of fostering an environment that encourages an on-going RTW negotiation process. Fostering such an environment is likely to aid effective communication of constraints between line managers and employees, benefiting both the implementation of appropriate work accommodations and positive RTW outcomes.

##### Hindrances to Re-Negotiation

The theme hindrances to re-negotiation describes the unsupportive experiences of participants owing to a toxic workplace culture that impeded the re-negotiation of appropriate accommodations, in turn negatively impacting sustainable RTW outcomes. Participants’ perceptions of lack of support during the RTW process appeared to be focused on the conflicts and tensions among colleagues and line managers. Participants who felt neglected, ignored, and unwelcomed on return, reported that it contributed to a decline in their health. According to participant 011-F-40+, because her colleagues and line manager were not of much help during her initial return, she was unable to negotiate necessary support, which resulted in a recurrent absence episode. Participant experiences, therefore, illustrate how line managers are instrumental to the workplace culture and how an environment the encourages toxic behaviours can hinder the re-negotiation process and overall RTW outcomes for employees who are the least powerful actors in the RTW negotiation process.

For these participants, unsupportive work cultures took the form of a lack of communication, poor reception on return to work, feelings of isolation, workplace conflict and stigmatisation/discrimination.


*“But being not so integrated in the team that is something I just accept that. I find it sometimes painful, allowing me feel what I feel, but I feel like I can’t change it really…”*
(015-F-40+, CMDs)


*“.... The flip side of that is this glass ceiling and you’re perceived to be a bit flaky. So how do you minimise that? Because you’ve had a bleep (mental break-down), that you’re a bit flaky and that you can’t do your job.”*
(011-F-40+, CMDs)

When employees are expected to return to perceived toxic environments, it increases the likelihood of a failed return, especially among people with CMD. According to participant 019-M-40+, his anxiety and depression were aggravated as a result of having his then-manager, whom he had grievances with, handle his RTW process.


*“It’s difficult because as I said then, going back to this previous line manager and you know after I got back last time and then dealing with this person who I could have put in a complaint with. So, it’s basically being line-managed by the person who was the problem and so I suppose it was an unusual case in that respect.”*
(019-M-30+, CMDs)

Perceptions of discrimination/stigmatisation, isolation and workplace conflict were raised among nine out of the twelve participants sick-listed with CMD, therefore, stressing the importance of normalising mental health conversations in the workplace to enable transition back to work and re-negotiation of adequate accommodations. The data also highlight the impact of line manager-worker relationships in either enabling or hindering their negotiation process: Line managers who have on-going conflicts with returning workers may not be the best placed to handle the RTW process.

The positive impact of a supportive work culture on the re-negotiation process was strengthened by the accounts of participants in other departments within the same organization, therefore providing an explanation for the inconsistencies in RTW outcomes across the organization.


*“In my experience with what I’ve had, it was very good. I think that it’s unique to me. I’m almost thankful for what I’ve got because I recognise that this isn’t standard and I don’t see it elsewhere within the organisation or … it’s in places, I mean there are pockets of really good behaviours. But you see other working environments, other businesses, you know your colleague work at places, and everyone is under a lot of pressure. I don’t see this across there.”*
(020-M-40+, CMD/MSD)

### 3.3. The Alignment of Key Stakeholders Informs the Choices Line Managers Are Presented with during the On-Going Re-Negotiation Process

This theoretical abstraction highlights the inter-relationship between themes relating to having a supportive line manager, the on-going dialogue/re-negotiation process, and the alignment of key stakeholders in facilitating the implementation of effective RTW strategies for returning employees which precedes sustainable RTW outcomes.

Participants suggest that the competence of managers is contingent on the level of understanding of the employee’s condition and its wider impact (on-going dialogue), which is mostly influenced by support from other services in implementing a suitable strategy (alignment of stakeholders). This, they say, stirs empathy on the part of line managers and influences their ability to effect beneficial strategies in a supportive capacity (choices presented to line managers) in the on-going re-negotiation process.


*“…it obviously depends on what your managers have to deal with if you like, …. because they have the Occupational Health and the HR and their guidance and obviously the HR team and the Wellbeing team would have dealt with a lot more situations with people’s mental health situations. I think they’re supported by the other members of the organisation, so they are able to support you. Even if they might not understand your situation that they haven’t dealt with any mental health issues themselves, I think they’re supported enough that they can be empathetic.”*
(021-F-40+)

### 3.4. Summary of Consolidated Theories

Overall, theory 1, the impact of alignment of key stakeholders around RTW strategies and a competent and supportive line manager on sustainable RTW outcomes, is justified in the accounts of these participants (see Table 1).

Therefore, strengthening the theoretical abstraction that sustainable RTW is not just facilitated by a supportive line manager, but also requires collaboration with key RTW stakeholders such as Occupational Health, GPs, HR to inform decisions around RTW strategies tailored to returning workers. Going beyond existing research, this study has identified that it is not as simple as having a supportive line manager alongside effective treatment, it is the choices that line managers are presented with that determines how supportive they can be as well as their motivation to be supportive.

Furthermore, an on-going process of negotiating workplace accommodations is important because with MSDs and CMDs, capacity fluctuates from day to day and recovery is not monotonic or stable. However, work environments and line managers that create conflicts and tensions have a high likelihood of hindering re-negotiation dialogues.

Therefore, participants’ accounts justify theory 2 and theory 3 shown in Table 2 and Table 3, respectively.

Overall, this study finds that involving the right people (worker, line manager and relevant support services) in the re-negotiation dialogue with the support of senior management and fostering a supportive workplace culture has the tendency to increase line manager’ capacity and produce better RTW outcomes for workers sick-listed with CMD and MSDs.

## 4. Discussion

The aim of this study was to provide insights into and understanding around the components of workplace RTW strategies that either facilitate or impede a sustainable RTW for employees with CMDs and/or MSDs sick-listed on either a long-term or short-term basis. The analyses have focused on explaining the link between the mechanisms and context that facilitate sustainable RTW outcomes for returning employees. These explanations were therefore drawn from three theoretical abstractions depicted in Figure 1, which forms one of the main contributions. Another contribution to the RTW literature from this study is the evidence clearly outlining the specific components of the good quality RTW strategies that are effective in facilitating a sustainable RTW for employees’ sick-listed on either short-term or long-term basis.

Our first theoretical abstraction highlights how the implementation of good RTW processes is contingent on the choices line managers are presented with, which could either be enabling or restricting. The positive effects of a good quality workplace RTW strategy in facilitating a sustained RTW is consistent with the broader RTW literature [22,32]. Like previous studies [6], we found that line managers play a role in either facilitating or impeding RTW outcomes. However, in addition, we found that line managers’ role extend beyond supportive line management to working in alignment with key stakeholders (occupational health, wellbeing team, HR, GP, etc.) around appropriate RTW strategies, thus, strengthening Cancelliere et al. (2016) and Corbière et al.’s [21] assertion that the RTW process is complex and involves the coordination of multiple stakeholders. Corbière et al.’s [21] study suggested the importance of line managers maintaining a working alliance with all RTW stakeholders. Our study, therefore, highlights the benefits of such alliances during the RTW process. In our study, this alliance with key stakeholders meant that line managers were privy to information spanning beyond helping returning employees get through the door to understanding employee’s ill-health, restrictions, and suggestions on appropriate RTW strategies to enable individuals perform and advance in their role on RTW. The effectiveness of this collaborative approach to RTW management for sick-listed employees is consistent with findings from previous studies [53]. Where line managers do not work with these stakeholders, they are restricted in their ability to effectively support employees. Moreover, accounts of participants revealed that where agreed RTW strategies were not approved by senior management, line managers were also restricted in their choices and supportive options, thereby negatively impacting employee’s sustained RTW outcomes. Our findings show that the effectiveness of RTW strategies hinges on senior management’s approval of agreed accommodations, therefore strengthening Phillips et al.’s [54] assertion of the importance of upper management’s commitment to accommodating employees with disabilities. Hence, we propose that policy makers make guidance provisions for leaders at all levels of organisations. A guidance that reiterates senior management’s duty of care to supporting workers with disability, clearly outlines the collaborating and supportive roles of line managers with other RTW stakeholders and caters to the training needs of line managers to enable their competencies in effectively managing the RTW process.

While our findings supports Baril et al.’s [55] suggestion that the success of RTW programs hinges on management’s commitment to the health and safety of its workers, our first contribution is that it is more than just about the commitment, but also about prevailing health and wellbeing logics in organisations—that is beliefs held in organisations about why health and wellbeing are important, with senior managers playing a critical role in influencing those logics that become adopted by other stakeholders through their symbolic actions and political actions [56]. For example, where senior management is committed to fostering a supportive workplace culture by way of defined policies, practices, or resources, it is reflected in the attitudes and behaviours of the workforce, thereby promoting the health and wellbeing of workers within that organisation. Hence, establishing the right or wrong kind of health and wellbeing logics may be a key contextual factor that influences the on-going adaptation or not of health and wellbeing practices, which appears to be the critical factor differentiating the success or not of such practices [57].

The second theoretical abstraction highlights that the RTW re-negotiation process appeared to yield favourable outcomes for returning employees when the workplace encouraged an on-going dialogue between line managers and employees on specific work concerns or issues. These dialogues broadly covered issues around work accommodation and workload expectations on RTW. On work accommodations adopted for returning employees, the decision to implement either a phased return or other flexible working options outside a phased structure was informed by employee’s medical restrictions and duration of sickness absence (short-term or long-term). We found that taking account of employee’s ill-health and the length of absence impacts the effectiveness of agreed RTW strategies. Our second contribution shows that the effectiveness of a phased return strategy or other flexible working options is dependent on the specific work component phased or flexible work accommodations provided for people based on their duration of absence. As employees classed as short-term absentees benefitted from flexible working options, those classed as long-term absentees benefitted from a phased RTW plan. Components of flexible working options provided included working from home, a few days off within the week, light duties (i.e., less demanding tasks) and half days within a full-time working contract until recovery. While components phased within the phased return strategy included reduced hours, reduced days, reduced workload, change in job role or level. Accounts of participants, therefore, showed that taking account of the length of absence of employees on RTW impacts the effectiveness of agreed strategies.

Our findings extend recent evidence showing the benefits of providing alternative work for employees with short-term disability [58], by detailing the strategies that benefit long-term absentees, who are linked to higher risk of disability and unfavourable RTW chances [58]. Conversely, like Lederer et al.’s [59], our study drew links between workload and negative RTW outcomes for returning workers. However, our third contribution extends Lederer et al.’s work to show that where RTW dialogues around workload between line managers and returning employee are not continuous, employees can be left managing inappropriate workloads, despite ill-health restrictions, which has detrimental effects on both health and RTW outcomes. This indicates the importance of having an on-going re-negotiation dialogue between line managers and employees throughout the RTW process, to identify and make reasonable adjustments for employees’ capacities at that point in time. We therefore propose that coordinators of RTW processes should ensure that RTW conversations, even after a return, are on-going. This can be achieved through agreed check-ins with worker to ensure fixed strategies are working, flag challenges for adjustments and provide continuous support which will in turn prevent a relapse and facilitate the sustainability of return.

A toxic workplace was also highlighted as a restrictive factor to the re-negotiation process between line managers and employees. While there is a growing literature on the negative impact of toxic workplace culture [60], our study is the first study to explore in depth the direct effects of a toxic workplace culture on RTW outcomes. In this study, toxic workplace culture was particularly perceived among people with CMDs as to how they were made to feel and the quality of interpersonal relationships among colleagues and line managers during the RTW process. Some participants found their work environment to be respectful and supportive, yet others experienced issues around isolation, conflicts, discrimination, and stigmatisation, which restricted the re-negotiation process and impacted negatively on their RTW outcomes. The restriction of the re-negotiation dialogue was particularly heightened in situations where employees had on-going conflicts with their line manager, therefore influencing the quality of RTW strategies agreed and employees’ overall RTW outcomes. Our findings suggest that fostering a supportive workplace that encourages a continuous effective communication during the RTW process has the likelihood to enable the re-negotiation dialogue and improve the quality of agreed work accommodations that would facilitate a sustainable RTW. While it may be difficult to manage people’s varying perceived perceptions of the environment and people they work with, we believe that employers would see better engagement and outcomes if returning workers are empowered with some level of control on who should manage their return-to-work process.

Finally, our third theoretical abstraction strengthens some author’s suggestion that a sustained RTW is facilitated by an interplay of multiple factors [6,22]. In our study, the RTW process which facilitated sustainable RTW outcomes for employees appeared to be a dynamic process involving an interplay of the mechanisms highlighted in both theoretical abstractions previously discussed. This finding, which is our fourth contribution, goes beyond existing research to show that helping sick-listed employees’ RTW is not as simple as just having a competent and supportive line manager. Rather, it requires line managers working with different stakeholders with different information and powerbase, alongside the employee, being the least powerful, which equips managers and employees to effectively manage the on-going re-negotiation dialogues on appropriate RTW strategies. Because it has been established that there is no one-size-fit-all approach to RTW, we propose that alongside strengthening line manager’s capacity by granting them access to key RTW/rehabilitation professionals, ensuring returning workers are in the centre of that decision conversation is important. They understand their conditions best, they know where they are in their recover at the point of return, they know aspects of their job they can and cannot do, and therefore, are in the best position to suggest how best to support their return to work. In this way, agreed RTW strategies will be tailored to individuals, thereby saving the cost of expecting employers to develop generic approaches that do not cater to the needs of workers with varying conditions.

### 4.1. Limitations of the Study

The selection and interview process were aimed at being thorough and rigorous to aid ease of applicability and transferability. However, a few limitations were identified in this study. While a limited number of 22 participants were recruited for this study, repeated interviews with these same participants provided sufficient data for triangulation of accounts. Hence, concurrent themes across participants in this study strengthened conclusions drawn [61].

Having more female participants (15) compared to male (7) could raise arguments about the accuracy of findings in this study. However, evidence show that more women than men in the public sector are likely to be on sick leave, therefore, shedding light on the challenges we faced in recruiting sick-listed male participants for the study [62]. As a result, the likelihood of recruitment bias in this study was heightened. However, by extending recruitment into two organisations, which aided the validity of findings and ease of generalisation within the public sector setting, this was mitigated.

In the second round of interviews, alongside face-to-face interviews, three telephone interviews were conducted with three participants based on request. It is possible that this could have introduced a certain level of respondent bias. However, this was mitigated using triangulation which took the form of validating interpretations across participants. According to [63], the absence of visual cues via telephone has a high chance of resulting in loss of contextual and non-verbal data and could compromise rapport, probing and interpretation of accounts. However, loss of such data due to absence of visual cue in telephone interviews did not apply to this study as repeated interviews were conducted for the purposed of clarifying ideas already generated in the first interviews. Henceforth, a telephone interview was considered acceptable for second interviews. Additionally, losing two participants in the second interviews due to resignation did not impact the reliability of interpretation drawn from accounts provided in the first interview. However, further triangulation was conducted by way of continually comparing generated data within and across case to enhance interpretive reliability.

### 4.2. Implications for Policy, Practice, and Research

Organisational, managerial, job and individual factors shape the RTW experiences of employees. Additionally, the relationship of the employee with direct line manager is the primary arena for provision of RTW flexibility, understanding and role adjustments [15]. Thus, of critical importance to good RTW practices is the alignment of line management, senior management and health services, and consideration of employees’ capacity to work at a given level [15]. It is also imperative to address the stigmatisation of mental health issues in the workplace by normalising the conversations around people’s experiences, as this will allow for better understanding and support.

Furthermore, vital to sustainable RTW is ensuring that the organisation and its processes are fit for purpose, i.e., there are no conflicts or that any conflicts can be accommodated if not resolved. The implication for research, therefore, is to investigate how employers can align line manager behaviours, senior manager decisions and interactions with health services, because it would be a dynamic process involving different stakeholders each with access to different information and power bases. The potential for conflict in respect of employee health has been identified as an area where more enquiry is needed [57].

## 5. Conclusions

Our study has provided a significant contribution to an otherwise limited knowledge base around the interactive role of crucial workplace factors on a sustainable RTW after ill-health. Findings from this work emphasise the benefits of involving key RTW stakeholders and returning employees in the RTW conversation, alongside the support of upper management to increase the capacities and autonomy of line managers in better supporting workers back to work. The importance of fostering a supportive workplace culture cannot be overemphasised as it has direct and indirect effects on the enabling or hindering choices of both employees and line managers during the re-negotiation process.

## Figures and Tables

**Figure 1 ijerph-20-01057-f001:**
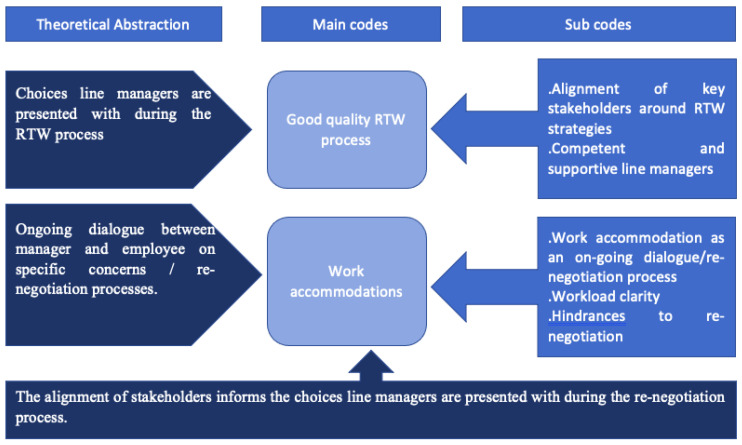
Final coding framework.

**Table 1 ijerph-20-01057-t001:** Initial theory.

	CMO RTW Theories	Original Theme
1	A competent and supportive manager, working in collaboration with other health services within the organisation (context) is likely to increase their level of understanding about employee’s condition and best RTW approach to adopt, as well as be more empathic towards employees (mechanism). As a result, they can successfully implement an effective RTW strategy (mechanism) approved by senior management, thus impacting on sustainable RTW (outcome).	Good quality RTW process

**Table 2 ijerph-20-01057-t002:** Initial theory refined.

	CMO RTW Theory	Original Theme
2	Reassuring workers of their workload during the on-going RTW negotiation process for appropriate work accommodation(context) is effective in assuaging fear (mechanism) and assisting in easy transition back to work (outcome), which in turn impacts on successful RTW (outcome).	Workload clarity

**Table 3 ijerph-20-01057-t003:** New theory developed.

	CMO RTW Theory	New Theme
3	When employees sick listed with CMD return to toxic working environments (context) during the RTW process (mechanism), it is likely to impede the negotiation process for adequate work accommodation, thereby aggravating their condition, leading to a failed RTW (outcome)	Hindrances to re-negotiation

## Data Availability

The data presented in this study are available within the article and the Appendix A here.

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
