# Peer review of "Sustainable Return to Work for Workers with Mental Health and Musculoskeletal Conditions"

_ijerph, 2023, doi:10.3390/ijerph20021057_

Round 1

Reviewer 1 Report

Congratulation authors.  This is a very good and interesting articles.  However, I would like to give some suggestions that you can consider in improving the articles.

1. Data collection - this section is written in detail however I would suggest the authors to add the evaluation Phase so that it may ease the reading.  Eg. Phase 1-theory gleaning and continue with the explanation on the process, then Phase 2 - refining or creation and Phase 3 -Theory consolidation as highlighted in Table S2.  

2.  The discussion is clearly written.  The authors can add value to the discussion by adding propositions for each of the theoretical abstraction.

3.  Section 4.1, please add limitation of this study.

Cheers. 

Author Response

Thank you for your positive comments on our paper. We hope the IJERPH readers find it just as interesting and useful. We are appreciative of the areas of improvement you have suggested, and we have addressed each suggestion (please the attached). We were very conscious of the word count and as such were unable to include every detail in the paper.

Reviewer 2 Report

Need correction in line 419.

The work was well structured and with a methodological approach adequate to the objectives.

The results are interesting to start a quantitative research on the subject as they present structured hypotheses so that progress can be made.

Author Response

Thank you for your comments on the adequacy of the methodological approach adopted for this study and the areas of further research the findings lend itself to. We are appreciative of the error you pointed out on line 419. That has now been addressed. Figure 1 was not properly cross-referenced, which generated the error. This has now been properly cross referenced as highlighted in the extract below.

Reviewer 3 Report

This is a review of the manuscript titled "Sustainable return to work for workers with mental health and musculoskeletal conditions". This qualitative study examined the return to work experiences of employees who suffered from common mental health and musculoskeletal disorders. Two main codes and five subcodes were identified and categorized into three theoretical abstractions. This manuscript could be improved if the following concerns are addressed:

1. I suggest the authors define and describe common mental disorders and musculoskeletal disorders in more detail.

2. It would be better to explain in more detail why this study focused on employees who suffered from common mental disorders and musculoskeletal disorders. In the literature on return to work, studies have focused on employees with other physical or mental health problems.

3. I recommend the authors to describe the three theoretical abstractions in more detail in the Introduction section.

4. The results of analyses could be improved if the differences in the return to work experiences among those with common mental health orders, those with musculoskeletal disorders, and those with both disorders.

5. More limitations of this study should be discussed.

Author Response

Thank you for the suggestions you kindly provided to improve our paper. We hope that we have been able to satisfy your concerns with the amendments made. The attached shows how we have addressed each suggestion.

Round 2

Reviewer 3 Report

The manuscript has been improved and is acceptable for publication in the current form.